# Explainable COVID-19 Detection Based on Chest X-rays Using an End-to-End RegNet Architecture

**DOI:** 10.3390/v15061327

**Published:** 2023-06-06

**Authors:** Mohamed Chetoui, Moulay A. Akhloufi, El Mostafa Bouattane, Joseph Abdulnour, Stephane Roux, Chantal D’Aoust Bernard

**Affiliations:** 1Perception, Robotics, and Intelligent Machines (PRIME), Department of Computer Science, Université de Moncton, Moncton, NB E1A 3E9, Canada; moulay.akhloufi@umoncton.ca; 2Montfort Academic Hospital, Institut du Savoir Montfort, Ottawa, ON 61350, Canada; mostafabouattane@montfort.on.ca (E.M.B.);

**Keywords:** RegNet, convolutional neural networks, COVID-19, deep learning

## Abstract

COVID-19,which is caused by the severe acute respiratory syndrome coronavirus 2 (SARS-CoV-2), is one of the worst pandemics in recent history. The identification of patients suspected to be infected with COVID-19 is becoming crucial to reduce its spread. We aimed to validate and test a deep learning model to detect COVID-19 based on chest X-rays. The recent deep convolutional neural network (CNN) RegNetX032 was adapted for detecting COVID-19 from chest X-ray (CXR) images using polymerase chain reaction (RT-PCR) as a reference. The model was customized and trained on five datasets containing more than 15,000 CXR images (including 4148COVID-19-positive cases) and then tested on 321 images (150 COVID-19-positive) from Montfort Hospital. Twenty percent of the data from the five datasets were used as validation data for hyperparameter optimization. Each CXR image was processed by the model to detect COVID-19. Multi-binary classifications were proposed, such as: COVID-19 vs. normal, COVID-19 + pneumonia vs. normal, and pneumonia vs. normal. The performance results were based on the area under the curve (AUC), sensitivity, and specificity. In addition, an explainability model was developed that demonstrated the high performance and high generalization degree of the proposed model in detecting and highlighting the signs of the disease. The fine-tuned RegNetX032 model achieved an overall accuracy score of 96.0%, with an AUC score of 99.1%. The model showed a superior sensitivity of 98.0% in detecting signs from CXR images of COVID-19 patients, and a specificity of 93.0% in detecting healthy CXR images. A second scenario compared COVID-19 + pneumonia vs. normal (healthy X-ray) patients. The model achieved an overall score of 99.1% (AUC) with a sensitivity of 96.0% and specificity of 93.0% on the Montfort dataset. For the validation set, the model achieved an average accuracy of 98.6%, an AUC score of 98.0%, a sensitivity of 98.0%, and a specificity of 96.0% for detection (COVID-19 patients vs. healthy patients). The second scenario compared COVID-19 + pneumonia vs. normal patients. The model achieved an overall score of 98.8% (AUC) with a sensitivity of 97.0% and a specificity of 96.0%. This robust deep learning model demonstrated excellent performance in detecting COVID-19 from chest X-rays. This model could be used to automate the detection of COVID-19 and improve decision making for patient triage and isolation in hospital settings. This could also be used as a complementary aid for radiologists or clinicians when differentiating to make smart decisions.

## 1. Introduction

Coronavirus disease 2019 (COVID-19) has been responsible for over 670 million cases and over 6.8 million deaths worldwide [1]. Real-time polymerase chain reaction (RT-PCR) is currently the gold standard for detecting and diagnosing severe acute respiratory syndrome coronavirus-2 (SARS-CoV-2) [2]. However, RT-PCR testing can still produce false-negative results [3]. Furthermore, the efficiency and timeliness of obtaining valid clinical results have become very important. The volume of patients requires the judicious use of resources while providing quality services and maintaining the safety of patients and healthcare professionals. As one of the most widely used diagnostic tools in medical practice, lung radiography adds undeniable clinical value in the diagnosis of many diseases [4]. The advantages of this artificial intelligence (AI)-based approach lie in its low cost; operational simplicity; and availability in a variety of clinical settings, both hospital- and community-based [4,5,6,7,8]. Although any clinician can obtain a clinical impression from an image of the lungs, radiography results must be validated by a radiologist. Thus, the implementation of this method in a high-volume diagnostic setting may be self-limiting; that is, the speed of validating the results depends on the availability of a radiologist and the volume of images to be reviewed [4,5,6,9,10,11]. Thus, the automatic detection of lung disease by AI is currently a highly valued and frequently evaluated concept in the fields of medical informatics research and radiology [4,12]. Several studies are already available. For the most part, deep learning approaches are applied to chest X-ray (CXR) images to classify COVID-19-infected patients, and the results have been shown to be very good in terms of accuracy (ACC), area under the curve (AUC), sensitivity (SN), and specificity (SP).

## 2. Related Work

Akinyelu et al. [13] introduced deep learning (DL)-based solutions for COVID-19 diagnosis using computerized tomography (CT) scans and various convolutional neural network (CNN) models. The authors used 9000 COVID-19 and 5000 normal images. All the CNN models were pre-trained. The findings showed that NASNetLarge [14], InceptionResNetV2 [15], and DenseNet169 [16] achieved the highest classification accuracy. The accuracy of the three models was 99.8%, 99.7%, and 99.7%, respectively. Khalil et al. [17] presented a pre-trained CNN called EfficientnetB4 [18]. They developed an in-depth training approach to extract the features of COVID-19 after a medical assessment before infection testing. The proposed framework achieved an accuracy of 97.0%. Hasan et al. [19] proposed a CNN called CVR-Net for COVID-19 diagnosis. The proposed end-to-end CVR-Net was an ensemble model with multiple scales and multiple encoders that combined the outputs from two separate encoders and their various scales to represent the final prediction probability. Their approach achieved accuracy scores of 99.8%, 98.4%, and 88.7% in binary classification for three-class and four-class classification. Abdul et al. [20] presented a deep learning multi-layered network to classify CXR images as COVID-19-positive or -negative. The proposed CNN used a dataset of patients infected with Coronavirus, wherein specialists indicated multi-lobar involvements in the CXR images. The authors used a total of 6,500 CXR images for model development. Their CNN model achieved an accuracy of 94.0%. Sahlol et al. [21] created a classification strategy by merging a pre-trained CNN (inception) and swarm-based feature selection method (fractional-order marine predators algorithm) to detect COVID-19 from CXR images. The developed method was assessed on two different datasets acquired from separate sources. Dataset 1 included 1675 non-COVID-19 samples taken from the Kaggle dataset [22] and 200 COVID-19 images acquired by Cohen, Morrison, and Dao [23]. Researchers from the University of Qatar and the University of Dhaka and fellows from Malaysia and Pakistan contributed to dtaset 2 [24]. In dataset 2, which consisted of 219 COVID-19 and 1341 non-COVID-19 CXR images, some positive COVID-19 samples from the SIRM dataset were added. The authors achieved accuracy scores of 98.7% and 98.2% for dataset 1 and dataset 2, respectively. Kumar et al. [25] proposed DL network called “LiteCovidNet” to detect COVID-19 cases as the binary class (COVID-19 vs. normal) and the multi-class (COVID-19 vs. normal and pneumonia) using CXR images. Their method achieved an accuracy of 100% and 98.82% for binary and multi-class classification, respectively. Muhammad et al. [26] fine-tuned a pre-trained model with some extra CNN layers (average pooling layer and two dense layers followed by ReLU with a softmax activation function). The authors used CXR images for binary classification (COVID-19 vs. negative). They benchmarked various CNN models such as VGG19 [27], Xception [28], ResNet152 [29], ResNet152v2, ResNet101, ResNet101v2, DenseNet201 [16], DenseNet169, and DenseNet121. Their best model achieved an average accuracy score of 95.0%. Ayalew et al. [30] presented a hybrid approach combining a convolutional neural network (CNN) and a histogram of oriented gradients (HOG) called DCCNet for COVID-19 diagnosis using CXR images. Their hybrid model achieved an accuracy score of 99.67%. Ghose et al. [31] presented transfer learning for COVID-19 detection using CT scans and CXR images. The authors merged CT scans with CXR images to create a global dataset. Their algorithm obtained an accuracy score of 99.59% for CXR and 99.95% for CT scan images.

Indumathi et al. [32] presented a method based on a machine learning (ML) algorithm to identify the degree of infection of COVID-19. The ML algorithm classified COVID-19-affected regions into various zones such as danger, moderate, and safe zones. Their proposed approach obtained an accuracy score of 98.06%. Salau et al. [33] provided a support vector machine (SVM) algorithm for the identification and classification of COVID-19. The authors used a discrete wavelet transform (DWT) algorithm for feature extraction and SVM for classification. Their method achieved an accuracy score of 98.2%.

Frimpong et al. [34] presented an interesting study on COVID-19 detection based on a Wi-Fi-enabled microcontroller, a temperature sensor, and a heart rate sensor. The authors designed a low-cost hardware system for students. The suggested method monitored the student’s condition continuously on a mobile application while detecting and differentiating between normal and abnormal body temperatures and regular and irregular heartbeats. Tests over time demonstrated the IoT-enabled system’s dependability, responsiveness, and affordability. The microcontroller’s intelligent programming and the sensor’s operation through the mobile application enabled the low-cost early diagnosis of abnormal temperature and heartbeat anomalies.

Lua et al. [35] presented a multi-scale class residual attention (MCRA) network for the multi-class classification of COVID-19, pneumonia, and normal cases using CXR images. The authors used the pixel-level image mixing of local regions for data augmentation and noise reduction. Their experimental results showed that their network achieved an accuracy score of 97.71%. Chouat et al. [36] presented a series of pre-trained DL models, ResNet50, InceptionV3, VGGNet-19, and Xception, for COVID-19 detection on CXR and CT scan images. The authors included a data augmentation technique to increase the size of the dataset. They found that VGGNet-19 outperformed the other three DL models on the CT image dataset, where it achieved an accuracy score of 87.0%. The best model for CXR images was Xception, with an accuracy score of 98.0%. Deriba et al. [37] presented three ML algorithms, naïve Bayes (NB), artificial neural network (ANN), and SVM, for COVID-19 detection. The authors used 311 patients’ data, comprising 214 males and 96 females. The model was tested using n = 10 input variables. The results demonstrated that the SVM algorithm achieved an accuracy score of 91.3%, and the other two methods provided an accuracy of 87.75% and 96.05%, respectively. A similar study presented by Wubineh et al. [38] for COVID-19 detection used a dataset of 1,048,575 variables obtained from Kaggle for model development. The authors employed a method called the PART rule-based algorithm and achieved an accuracy score of 92.47% using a 10-fold cross-validation test.

In this study, a CNN algorithm for COVID-19 detection was developed. A preliminary internal validation was carried out with a balanced cohort of patients from Italy, i.e., patients with an official diagnosis of COVID-19 and others with a negative or different diagnosis of COVID-19. The anonymized images of this cohort were obtained from the “Società Italiana di Radiologi Medica e Interventistica” [39]. The results of this first study showed a sensitivity of 98% and a specificity of 97%. However, the internal validation was carried out at a small scale, and the continuity of the model training on a larger scale had to be ensured as a validation process for its eventual clinical use. Thus, the objective on this study was to validate and test this deep learning model on confirmed cases to detect COVID-19 from chest X-ray (CXR) images. We aimed to make the following contributions:1.A state-of-the-art pre-trained CNN model called RegNetX032 was fine-tuned for multi-binary classification (COVID-19 vs. normal, COVID-19 + pneumonia vs. normal, and pneumonia vs. normal. Such a model has not yet been proposed in a medical imaging classification study. Our study investigated the performance of this fine-tuned RegNet model for COVID-19 detection.2.We used various datasets, which differed in terms of resolution quality, to validate the performance of the model and its degree of generalization.3.We tested the performance and the degree of generalization of the model using a private dataset.4.An explainability model was integrated to localize the signs of the disease and provide decision support.The paper is structured as follows: Section 1 introduces the COVID-19 pandemic, and Section 2 presents related work. Section 3 discusses the methods and presents the proposed deep learning model and the datasets used in this study. Section 4 presents the experimental results. Section 5 describes the model’s explainability. Section 6 discusses the model’s limitations. Section 7 presents a discussion and conclusions.

## 3. Methods

### 3.1. Deep Learning Model

For COVID-19 detection, we fine-tuned the recent convolutional neural (CNN) network called RegNetX032 [40]. Convolutional neural network architectures have often been created and optimized for a single objective. For instance, at the time of its original release, the ResNet [29] model family was tuned for accurate results on ImageNet [41]. MobileNets [42] were designed specifically to perform on mobile devices, as the name suggests. EfficientNet [43] was developed to be highly effective in visual recognition tasks.

Radosavovic et al. [40] decided to set a very unusual but extremely interesting goal in their study “Designing Network Design Spaces”. The authors set out to investigate and develop a highly flexible network architecture that was customizable for the best classification performance, could be developed to run on mobile devices or be extremely effective, and was also highly accurate. Setting the proper parameters in a quantized linear function, which is a sequence of formulas with specified parameters to determine a network’s width and depth, was thought to be able to manage this adaptation. They also used a novel method, creating a network named network design spaces rather than manually creating the model architecture.

### 3.2. Deriving the RegNet Model from Network Design Spaces

A network design space is made up of various model architectures, as the name might imply, but it also builds various parameters that create a space of alternative model designs. This is not like a neural architecture search, wherein the developers experiment with several structures to find the best one, adjusting, for example, the network’s width, depth, or groups. RegNet [40] also only employed one type of network block out for the several architectures, i.e., the bottleneck block. The authors first created a space for all practical models, which they referred to as “AnyNet”, before reaching the final RegNet design space. This part generated a large variety of models from a large variety of combinations of the different parameters. On the ImageNet dataset, all these models were trained and tested using a standard training phase (epochs, optimizer, weight decay, and learning rate scheduler). By examining the parameters that contributed to the improved performance of the best models in the AnyNet design space, they developed gradually smaller iterations of the original AnyNet design space. In general, they tested the weighting factors of several parameters to reduce the design space to only the best models. Setting a shared bottleneck ratio and a shared group width as well as parameterizing the width and depth to increase in the later stages were some of the enhancements applied from the existing design space to the tighter design space. They finally reached the optimized RegNet design space, which only showed the best models and the quantization linear function required to specify the models.

### 3.3. The RegNet Design Space

The network was constructed of several stages consisting of multiple blocks, forming a stem (start), body (main part), and head (end). There were different stages specified inside the body, and each stage was made up of different blocks. As previously mentioned, the standard residual bottleneck block with group convolution was the only type of block used in RegNet.

The RegNet model’s architecture was determined by a quantized linear function that was controlled by the selected parameters rather than by fixed parameters such as depth and width. After optimization, the following formula was used to determine the block widths:(1)$$u_{j}=w_{0}+w_{a}.j\;\;for\;0⩽j<d.$$

The width for each block increased by a factor of $w_{a}$ for each additional block. The authors then introduced an additional parameter $w_{0}$ (set by the user) and calculated sj:(2)$$w_{j}=w_{0}.w_{m}^{sj}.$$

Finally, the authors rounded sj and computed the quantized per-block widths in order to quantize $u_{j}$.

All blocks with the same width were simply counted together to form one stage to determine the width for each stage *i*, as all blocks combined should have the same width. The authors set the parameters *d* (depth), $w_{0}$ (initial width), $w_{a}$ (slope), $w_{m}$ (width parameter), *b* (bottleneck), and *g* (group) in order to generate a RegNet from the RegNet design space. The authors altered these settings in order to create various RegNets with diverse characteristics.

In this study, we used RegNetX032, which represents 3.2 billion flops. The reasons for choosing this version were that it is fast in terms of convergence and obtained a high accuracy score of 94% on the Imagenet [41] dataset. For each binary classification, we customized the pre-trained model by adding global average pooling, followed by batch normalization and two dense layers of sizes 512 and 128, respectively. To reduce overfitting, each dense layer was followed by a dropout layer (25%). Finally, a softmax layer provided the probability prediction scores for the multi-binary classification: (1) COVID-19 positive vs. normal cases; (2) COVID-19 + pneumonia vs normal cases; and (3) pneumonia vs. normal cases. Figure 1 provides an overview of our approach.

### 3.4. Patients and Datasets

#### 3.4.1. NIH Dataset

For the pneumonia and healthy classes, we used the NIH [44] chest X-ray dataset, comprising 112,120 CXR images with disease labels from 30,805 unique patients. This dataset was obtained from the National Institute of Health (USA). There were 15 classes in the dataset (14 diseases, and one class for “healthy”). Infiltration, edema, atelectasis, pneumothorax, consolidation, emphysema, effusion, fibrosis, pneumonia, cardiomegaly, pleural thickening, mass, nodule, and hernia were some of the available disease images. Expert physicians assigned grades to the CXR images. We reserved 6000 CXR images from the healthy category and 4852 for the other pneumonia cases (pneumothorax, effusion, etc.). We obtained a total of 10,852 images for the training and validation sets. Figure 2 shows example NIH CXR images.

#### 3.4.2. COVID-19 Image Data Collection

Cohen et al. [23] released an open dataset of CXR and CT scan images of patients who were positive for COVID-19 and other viral/bacterial forms of pneumonia (MERS, SARS, and ARDS). The data were mainly scraped from online medical websites collecting released COVID-19 images from hospitals and physicians. The dataset contained 654 COVID-19 CXR images, and its objective was to develop AI-based approaches to predict and understand the infection. Figure 3 shows examples from the COVID-19 image data collection dataset.

#### 3.4.3. COVID-19 Radiography

The database in [45] contains 219 CXR COVID-19-positive images collected by a team of researchers from the University of Qatar (Doha, Qatar) and the University of Dhaka (Bangladesh) and their Pakistani and Malaysian collaborators with the aid of various medical doctors, who created a CXR image database for positive cases of COVID-19. Figure 4 shows examples from the COVID-19 radiography dataset.

#### 3.4.4. BIMCV COVID19+

The BIMCV COVID19+ [46] dataset is a broad dataset of COVID-19 patients’ CXR and computed tomography (CT) images together with their radiographic observations, pathologies, polymerase chain reaction (PCR) test results, diagnostic antibody tests for immunoglobulin G (IgG) and immunoglobulin M (IgM), and radiographic records from the Medical Imaging Databank in the Valencia Area Medical Imaging Bank (BIMCV). The images were collected by a team of specialist radiologists in high resolution and annotated. In addition, comprehensive information was provided, including demographic information for the patient, projection type (PA-AP), and criteria of acquisition for imaging analysis. This database included 1380 customer experience (CX), 885 digital transformation (DX), and 163 CT images. The images were merged into a single dataset with a total of 4148 COVID-19-positive images and 10,852 images of healthy patients and pneumonia cases, providing a total of 15,000 CXR images. Figure 5 shows examples from the BIMCV COVID19+ dataset.

#### 3.4.5. Montfort Dataset

In addition to the above datasets, we collected more images in collaboration with health professionals from Montfort hospital (Ontario, Canada) and built the Montfort dataset for the testing phase. This proprietary dataset included 176 adults (~18 years of age and older) with a total of 236 CXR images. Of these, 93 patients (150 CXR images) were COVID-19-positive, as confirmed by positive RT-PCR test results and/or diagnosis by a physician for COVID-19. Added to the dataset were 26 patients with pneumonia (other than COVID-19, 29 CXR images) and 57 patients with healthy lungs (57 CXR images). These patients were labeled using radiology reports and RT-PCR tests. Figure 6 shows examples from the Montfort dataset.

All training was carried out using the Python 3.8 programming language with the Keras [47] library on a workstation running 8 Nvidia GeForce RTX 2080ti [48] cards (12 GB of RAM each). The batch size was held constant at eight for the fine-tuned model. The model was trained for 200 epochs, and all experiments used the Adam optimizer [49] with a learning rate of 1 × 10${}^{-3}$, which was further reduced when the validation accuracy did not improve consecutively over three epochs. We did not apply augmentation techniques, and CXR images were resized to 512 × 512.

## 4. Results

The performance of the model was calculated using the accuracy score and the receiver operating characteristic curve (ROC). The area under the ROC curve (AUC) was used as the measure of diagnostic accuracy for the model. A 0.5 threshold was used to validate the detection of a specific class. Furthermore, using the RT-PCR results as a reference for COVID-19 cases and radiology reports for pneumonia (other than COVID-19) and healthy cases, sensitivity and specificity were calculated. These measures were calculated as follows:(3)$$SN=\frac{TP}{TP+FN}$$
(4)$$SP=\frac{TN}{TN+FP}$$
(5)$$ACC=\frac{TP+TN}{TP+FN+TN+FP}$$
where TP is the true-positive rate, i.e., the number of positive cases that were correctly labeled; TN is the true-negative rate, representing the number of negative cases that were correctly labeled; FP is the false-positive rate, representing the number of positive cases that were falsely labeled; and FN is the false-negative rate. Three model scenarios were created comparing different conditions: scenario (1)—COVID-19 positive vs. healthy cases; scenario (2)—COVID-19 + pneumonia vs healthy cases; and scenario (3)—pneumonia vs. healthy cases.

The accuracy of the validation set for each model scenario was found to be 98.6%, 97.3%, and 95.0% for scenario 1, scenario 2, and scenario 3, respectively. Regarding the AUC scores, the models obtained values of 98.0% (scenario 1), 98.0% (scenario 2), and 97.0% (scenario 3). A value of 1.00 indicates a perfect COVID-19 and/or pneumonia test, and 0.50 (as plotted by the straight line of no discrimination) represents a diagnostic test that was no better than random coincidence. On the Montfort test set, the AUC for the model scenario showed better results with values of 99.1% (scenario 1), 99.1% (scenario 2), and 99.4% (scenario 3). The accuracy scores were found to be 96.0%, 95.3%, and 96.4% for scenario 1, scenario 2, and scenario 3, respectively.

Confusion matrices were constructed to summarize the binary classification performance of the model with the sensitivity and specificity (Figure 7) for the testing phases. The validation phase showed excellent sensitivity and specificity results for all three scenarios, ranging between 95.0% and 98.0% (Table 1). The testing phase showed close results to the validation phase with sensitivity and specificity ranging between 90.0% and 98.0% (Table 1).

Table 2 presents a comparison with machine learning and deep learning methods for COVID-19 detection. As one can see, our approach obtained the best scores compared to most of the studies presented.

Our model’s score was very close to that of Ayalew et al. [30]. In their study, the validation and test phases were taken from the same dataset, and data augmentation was applied. This could have provided biased results due to the similarity of the images from the training and testing sets. Moreover, the hybrid architecture could have increased the complexity of training compared to using a single model for feature extraction and classification. In addition, the authors combined feature extraction, detection, and segmentation from multiple models, which could have also created a delay in image inference. In the stuyd of Ghose et al. [31], the details of the dataset division were not provided, and no explainability model was developed in order to visualize the detected signs.

We also note that no study has validated the performance of its model on an independent dataset to test the degree of generalization and prevent bias. Most studies have tested their model on a test set that was reserved from the global dataset. This confirms that our model was robust in terms of detecting COVID-19 using an independent and unique dataset.

The proposed model improved upon our previous model [50]. The model was based on EfficientNet-B0 and obtained an AUC of 95.0%, an SP of 90.0%, and an SN of 97.0%. This indicated that the current proposed model was robust and able to detect COVID-19.

## 5. Model Explainability

To confirm how the model learned to detect COVID-19 signs, we developed an explainability model based on gradient-weighted class activation mapping (Grad-CAM) [51]. This approach was used to generate a visual description of the outcomes of the proposed CNN models. Grad-CAM uses any target’s gradients flowing into the final convolutional layer to generate a coarse map of localization highlighting important regions in the predictive image. Grad-CAM was applicable to our proposed CNN model without any architectural changes or re-training. The proposed technique combined Grad-CAM with fine-grained visualizations to create a high-resolution class-discriminative visualization. Figure 8 shows samples of true-positive cases of COVID-19 detected with our fine-tuned model. As one can see, the model efficiently localized the infected area on the lung. Figure 9 presents some of false-positive CXR images. The low quality and the text on the radiography images confused the model when localizing the important areas of the disease on the lung.

## 6. Model Limitations

Despite the results obtained from the proposed model, we found that the model provided some false-positive detection results. This was due to the poor quality of some images from the Montfort dataset. For example, as shown in Figure 9 rows 1 and 2, the model detected a normal case as COVID-19-positive, and in row 3, a normal case as pneumonia. The artefact and the noise created an obstacle for the model, which interpreted them as signs of pneumonia. In future work, we will test different strategies to improve the quality of the images.

## 7. Discussion and Conclusions

Our study demonstrated that transfer learning can be effective in detecting COVID-19 using CXR images. Our pre-trained ImageNet model achieved a high sensitivity of 98.0% in detecting COVID-19-positive patients compared to healthy ones, and it demonstrated state-of-the-art performance in all measures discussed. This high performance ensured accurate diagnosis in most cases, even with limited data, which is typical in real-world situations. We also used the Grad-CAM visualization technique to make the proposed deep learning model more interpretable and explainable, which validated its performance and aided in the development of novel visual indicators for manual screening. However, there are still several research questions that need to be addressed. For instance, we need to focus on determining the severity of COVID-19 and developing robust models that can extract more features from CXR images to improve detection performance. Additionally, explanatory analyses could help us gain more insight into the mechanisms behind COVID-19 detection. Furthermore, it would be interesting to investigate whether our model could be applied to other respiratory diseases and explore the potential of transfer learning in diagnosing such diseases. Overall, our study provides a solid foundation for future research in this field. In conclusion, our study demonstrated that our algorithm, validated using CXR images from a large dataset with varying image quality and from different healthcare systems around the world, could provide greater imaging insights and a quantifiable probability of COVID-19 diagnosis compared to other respiratory diagnoses. The high performance of our algorithm could be useful in triaging patients for isolation in a timely manner and improving patient flow while waiting for other gold-standard testing results. The explainability of the images provides crucial information to assess lung damage and valuable insight for timely treatment and intervention. Our model could serve as a complementary aid in helping radiologists perform diagnoses and could potentially automate radiology services with AI-powered decision support tools. In the future, further research can focus on developing more robust models that can extract more features from CXR images to improve the performance of detection and investigate the application of transfer learning in diagnosing other respiratory diseases.

## Figures and Tables

**Figure 1 viruses-15-01327-f001:**
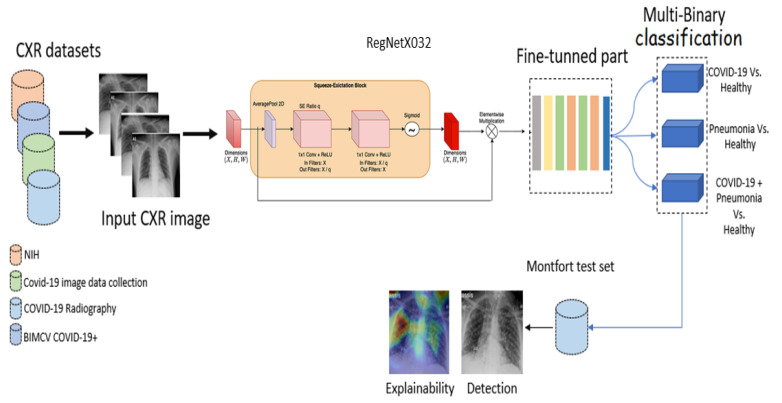
Proposed deep CNN architecture for COVID-19 detection.

**Figure 2 viruses-15-01327-f002:**
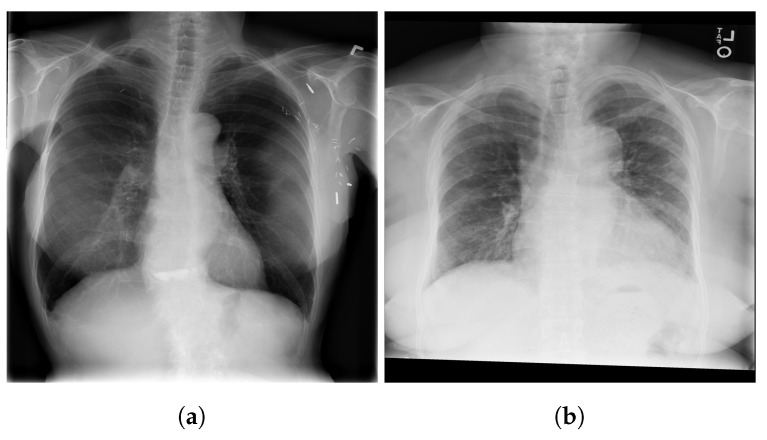
Examples of NIH CXR images; normal (**a**), pneumonia (**b**).

**Figure 3 viruses-15-01327-f003:**
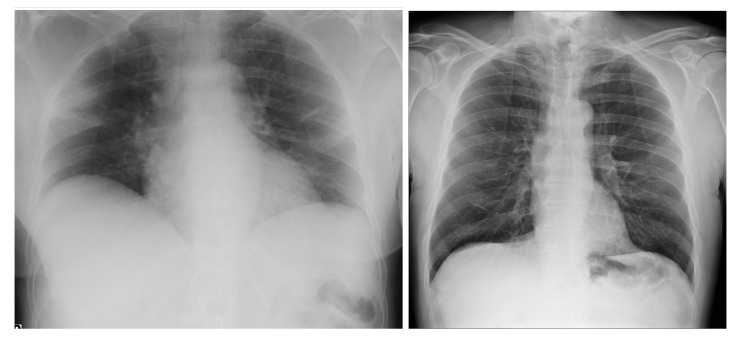
Examples of CXR images from the COVID-19 image data collection dataset.

**Figure 4 viruses-15-01327-f004:**
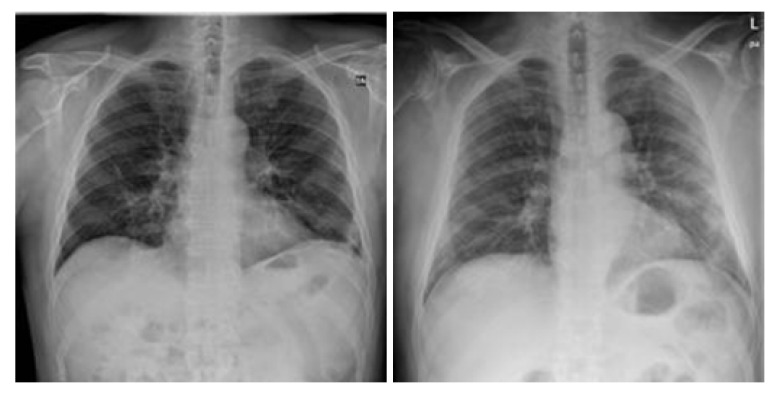
Examples of CXR images from the COVID-19 radiography dataset.

**Figure 5 viruses-15-01327-f005:**
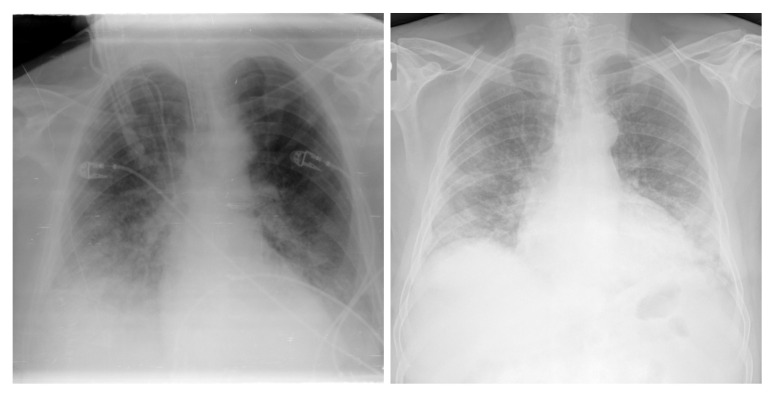
Examples of CXR images from BIMCV COVID19+.

**Figure 6 viruses-15-01327-f006:**
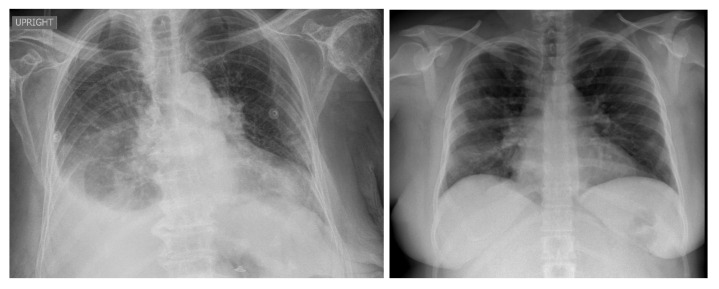
Example of CXR images from the Montfort dataset.

**Figure 7 viruses-15-01327-f007:**
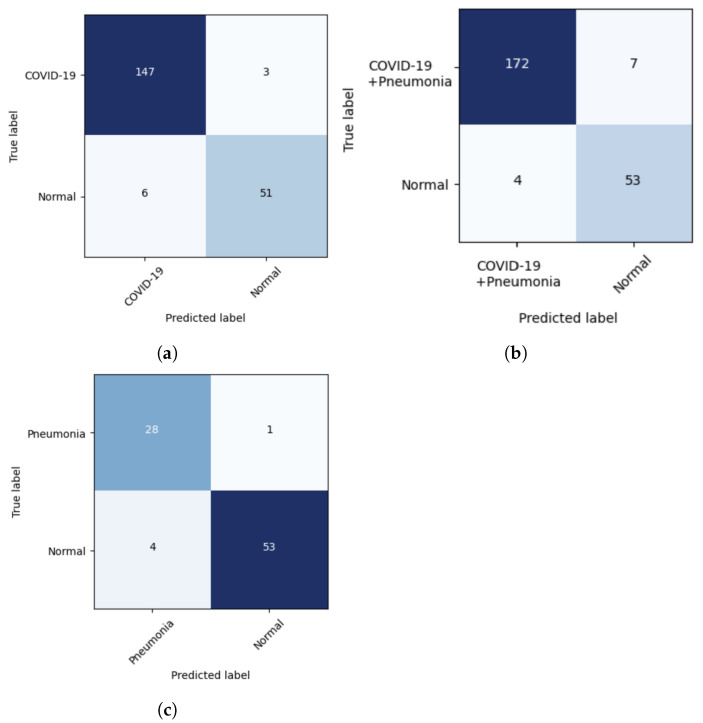
(**a**–**c**) Confusion matrix of the deep learning model for the three scenarios of the testing phase.

**Figure 8 viruses-15-01327-f008:**
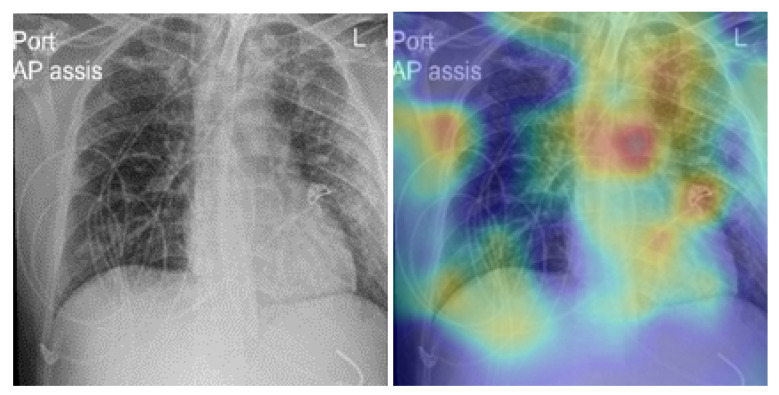
Explainabilty of true-positive cases of COVID-19. The green and yellow/red colors highlight important areas detected by the fine-tuned deep learning model (RegNetX032) on the CXR images.

**Figure 9 viruses-15-01327-f009:**
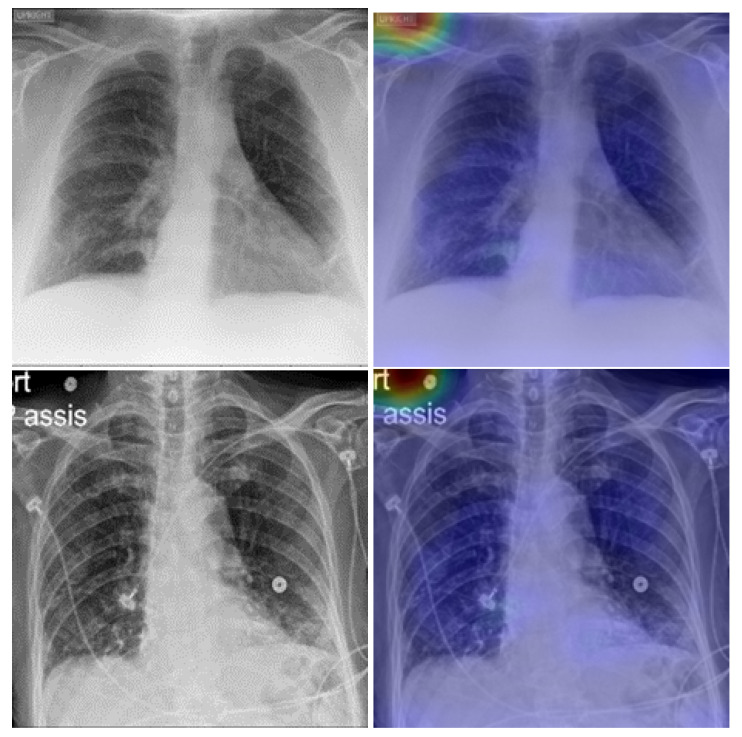
Explainabilty of false-positive cases of healthy patients. Healthy identified as COVID-19-postivie (rows 1 and 2); healthy identified as having pneumonia (row 3).

**Table 1 viruses-15-01327-t001:** Sensitivity and specificity for the three scenarios in the validation and testing phases.

	Scenario 1			Scenario 2			Scenario 3		
	ACC	SN	SP	ACC	SN	SP	ACC	SN	SP
Val (merged sets)	98.6%	98.0%	96.0%	97.3%	97.0%	96.0%	95.0%	95.0%	95.0%
Test (Montfort)	96.0%	98.0%	90.0%	95.3%	96.0%	93.0%	96.4%	97.0%	93.0%

**Table 2 viruses-15-01327-t002:** Performance comparison with state-of-the-art methods for COVID-19 detection.

Study	Method	ACC	AUC	SN	SP	COVID-19 Images	Explainibility
[30]	CNN and HOG	99.6%	-	-	-	3000 with data augmentation	No
[31]	CNN	99.5%	99.2%	99.5%	99.5%	1626	No
[32]	SVM	98.5%	-	88%	87.2%	250	No
[35]	CNN	97.71%	-	96.76%	96.56%	3338	Yes
[36]	4 CNN	98.0%	-	-	-	500	No
[17]	CNN	97.0%	-	-	-	217	No
[26]	CNN	94.5%	-	-	-	203	No
Our model	CNN	98.6% 96.0%	98.0% 99.1%	98.0% 98.0%	96.0% 96.0%	4148 150	Yes

## Data Availability

The data used in this work originated mainly from public datasets and a private dataset. Please see the section describing the datasets.

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
