# Peer review of "Explainable COVID-19 Detection Based on Chest X-rays Using an End-to-End RegNet Architecture"

_viruses, 2023, doi:10.3390/v15061327_

Round 1

Reviewer 1 Report (Previous Reviewer 3)

The authors have improved the paper. 

I will suggest the title be changed to this: Explainable COVID-19 Detection from Chest X-rays Using an End-to-End RegNet Architecture

The authors should proofread the paper once more to remove any possible errors.

Reviewer 2 Report (Previous Reviewer 1)

Dear Authors,

Current manuscript revision is fully mature and acceptable for publishing without further changes.

I warmly endorse its acceptance for publication.

Best Regards

This manuscript is a resubmission of an earlier submission. The following is a list of the peer review reports and author responses from that submission.

Round 1

Reviewer 1 Report

Dear Authors,

This is a worthy submission on deep learning applied in imaging diagnostics of an infectious disease.

It may actually fill certain knowledge gaps in the seminal literature.

However it has important bottle neck weakness.

its evidence base is overtly homogeneous too heavily relying on findings from wealthy OECD nations.

however the burden of corona to a large extent belongs to the LMICs nations of the Global South.

Thus in introductory part and discussion evidence base has to be diversified and expanded to include far more sources from these less wealthy regions.

Such change would increase transnational applicability of findings.

For this purpose I warmly recommend consideration for inclusion of at least several of the sources below alongside few additional ones at authros own disposal:

https://www.mdpi.com/2306-5354/8/6/84

https://www.tandfonline.com/doi/abs/10.1080/14737167.2020.1823221

https://www.tandfonline.com/doi/full/10.2147/RMHP.S341500

I remain willing to review the revised manuscript.

Author Response

Thank you for taking the time to review our paper, In the attachment the answers to your comments. 

Thanks

Reviewer 2 Report

Manuscript rewiev "Explainable COVID-19 Detection on Chest X-rays Using an End-to-End RegNet Architecture" by Mohamed Chetoui t al.

Major comment

1 - The authors state that "RT-PCR testing can still produce false negative results [3]. "In this manuscript (3) the authors demonstrated that RT-PCR testing can produce false negatives during the first day of symptom onset in where the probability of false negatives was 100%. It may be of interest to know on which day the chest X-ray was taken and to evaluate the effectiveness of the diagnostic image during the early stages of the infection. Therefore, the authors applied a method without verifying the real efficacy and assuming that the radiological diagnosis is discriminating. According to this approach, the different sets of data were built which then taught, evaluated and validated the effectiveness of the algorithm obtained. This approach, which assumes radiodiagnostic imaging as infallible, invalidates the rationale for the study that the authors set out to do. Let me give an example: if the RT-PCR classifies negative for COVID19 during the first day of illness and the X-ray is performed on the same day and it is also negative, this sample is then entered as negative in the training system, the algorithm created will have a high probability of carrying the same errors as RT-PCR.

2 - Covid19 does not always encounter pneumonia but it can proceed without necessarily affecting the respiratory tract and developing them. The RT-PCR performed in the laboratory and the radiographic study of the lungs are two diagnostic aspects of the sequelae that the infection can potentially cause in its course. The first test identifies the virus and the potential for clinical repercussions of the infection and the other instrument certifies the sometimes serious compromise of the respiratory system. The two tests are not mutually exclusive and neither can replace the other in the diagnostic process for the pathology determined by SARSCOV2.

3-  I agree with the authors that the advantage of this Artificial Intelligence (AI)-based approach lies in the low cost, simplicity of the process, and availability of the test in a variety of clinical settings, both hospital and community, but I would add that these systems should be built with rigid and standardized criteria.

Author Response

Thank you for taking the time to review our paper, In the attachment, we provide the point-by-point responses. 

Reviewer 3 Report

1.         The abstract is not detailed enough. Readers expect to see more detail of the methodology, results, and conclusion in the abstract. The abstract need to be greatly improved. The abstract does not show that the authors achieved much as there is no numerical justification to back the author’s claims or results of comparative analysis to show superior performance.

2.         In the introduction, the authors should explain why they did it (motivation) discussing the possible outcome. Readers are primarily interested in the motivation and outcome of your research. Therefore, a good introduction should contain:

a.         What is the problem to be solved?

b.         Are there any existing solutions?

c.         Which is the best?

d.         What is the main limitation of the best and existing approaches?

e.         What do you hope to change or propose to make it better?

f.          How is the paper structured?

3.         Please clearly highlight how your work advances the field from the present state of knowledge and you should provide a clear justification for your work which should be stated at the end of literature review/ related works. The impact or advancement of the work can also appear in the conclusion.

4.         The authors mentioned feature extraction but have not presented this stage in their work. A block diagram or flowchart of the steps would be helpful. The authors should look for recent works on feature extraction to cite. An example of such recent literature which the authors can consult amongst others is: Feature Extraction: A Survey of the Types, Techniques, Applications, 5th IEEE International Conference on Signal Processing and Communication (ICSC), Noida, India, pp. 158-164 (2019). DOI: 10.1109/ICSC45622.2019.8938371

5.         Related works section is not sufficient. The authors should improve on this section as they have left many papers out. Normally, it’s the gaps in work of others that the authors are expected to fill. Therefore, at the end of your review section state the problems in this field with appropriate reference and tell readers which one your work addresses.

-Explainable COVID-19 Detection on Chest X-rays Using an End-to-End Deep Convolutional Neural Network Architecture, https://www.mdpi.com/2504-2289/5/4/73

The authors should categorically state how their previous published paper cited above is different from the present.

The authors should consult and cite:

(i)                 Detection and Classification of COVID-19 Disease from X-ray Images Using Convolutional Neural Networks and Histogram of Oriented Gradients. Biomedical Signal Processing and Control, 103530, Vol. 74, pp. 1-11, 2022. DOI: 10.1016/j.bspc.2022.103530

(ii)               Detecting COVID-19 infection status from chest X-ray and CT scan via single transfer learning-driven approach, https://www.ncbi.nlm.nih.gov/pmc/articles/PMC9533058/

(ii)        Prediction of COVID-19 Outbreak with Current Substantiation Using Machine Learning Algorithms. Intelligent Interactive Multimedia Systems for e-Healthcare Applications. Springer, Singapore, 2022. https://doi.org/10.1007/978-981-16-6542-4_10

(iii)       Detection of Corona Virus Disease Using a Novel Machine Learning Approach. 2021 International Conference on Decision Aid Sciences and Application (DASA), pp. 587-590, 2021. DOI: 10.1109/DASA53625.2021.9682267.

(iii)             Innovative IoT-Based Wristlet for Early COVID-19 Detection and Monitoring Among Students. Mathematical Modelling of Engineering Problems, Vol. 9(6), pp. 1557-1564, 2022. DOI: 10.18280/mmep.090615

(iv)             COVID-19 diagnosis via chest X-ray image classification based on multiscale class residual attention, https://www.ncbi.nlm.nih.gov/pmc/articles/PMC9433340/

(v)               Artificial Intelligence Assisted Decision Making in Predicting COVID-19 Patient’s Path. Journal of Pharmaceutical Negative Results, Vol. 14(3), pp. 1250–1255, 2023. DOI: 10.47750/pnr.2023.14.03.166

(vi)             COVID-19 detection in CT and CXR images using deep learning models. Biogerontology 23, 65–84 (2022). https://doi.org/10.1007/s10522-021-09946-7

(vi) Knowledge Based Expert System for Diagnosis of COVID-19. Journal of Pharmaceutical Negative Results, Vol. 14(3), pp. 1242–1249, 2023. DOI: 10.47750/pnr.2023.14.03.165

(vii) Detection of Corona Virus Disease Using a Novel Machine Learning Approach. 2021 International Conference on Decision Aid Sciences and Application (DASA), pp. 587-590. DOI: 10.1109/DASA53625.2021.9682267.

6.         Most of the figures in this paper are not clear enough. The authors should endeavour to change them. E.g. Figs. 5 and 6, while Figs. 7, 8 should be enlarged.

7.         The authors need to discuss the results in Tables 3, 4, and 5 better. The reason why the proposed technique performs better has not been explained.

8.         There is no comparison of results with the existing works in this paper. This should be added for readers to see how your proposed method performs relative to other works.

9.         It would be good for the authors to discuss other methods that different authors have used to take COVID-19 Disease such as facemasks and hand washing solutions.

Authors should read and cite:

(i)         Low Cost Sensor Based Hand Washing Solution for COVID-19 Prevention. 2021 International Conference on Innovation and Intelligence for Informatics, Computing, and Technologies (3ICT), pp. 93-97, 2021. DOI: 10.1109/3ICT53449.2021.9581821.

(ii)        Facemask wearing to prevent COVID-19 transmission and associated factors among taxi drivers in Dessie City and Kombolcha Town, Ethiopia

10.       The authors should structure the paper into abstract, introduction, literature review/related works, methodology, results and discussion, and conclusion.

11.       I was hoping to see more results and discussion as more results could be presented to make the work much appreciable. The authors are encouraged to reduce the plagiarism of the paper.

12.       The Limitations of the proposed study need to be discussed before conclusion.

13.       Some of the challenges encountered during the course of the study can be highlighted and future recommendations can be added at the end of the conclusion. Retitle conclusion as conclusion and recommendation.

14.       The results and discussion section is very weak. The authors should endeavor to improve on this section. In the section of selection of local minima, what criteria did the authors used? Also what priors did the authors consider? What is the minimum and the maximum values? If these are suitable, do they work for different types of images or just the images under consideration?

15.       Lastly, no comparison of results was presented with other state-of-the-art methods which have used machine learning techniques. You can also add a comparison of the processing time.

16. The authors should add details of the computing device and justify its use.

17.       It’s quite unbelievable that the authors did not state the amount of data used for experimentation? Or what type of data was used (primary or secondary?).

18.       The authors should structure the paper into abstract, introduction, literature.

19. The authors can present the translators algorithm in tabular form. Also the computing systems specifications and capacity was not mentioned?

Author Response

(The authors gave the same response as above.)

Round 2

Reviewer 2 Report

no comment

Reviewer 3 Report

The authors have improved the paper. The authors should proofread the paper to remove all grammatical errors in the paper.